# Mr. HiSum: A Large-scale Dataset for Video Highlight Detection and Summarization

**Jinhwan Sul**[*]
Georgia Institute of Technology
jsul7@gatech.edu

**Jihoon Han**[*]
Seoul National University
joon7092@snu.ac.kr

**Joonseok Lee**[†]
Seoul National University
joonseok@snu.ac.kr

## Abstract

Video highlight detection is a task to automatically select the most engaging moments from a long video. This problem is highly challenging since it aims to learn a general way of finding highlights from a variety of videos in the real world. The task has an innate subjectivity because the definition of a highlight differs across individuals. Therefore, to detect consistent and meaningful highlights, prior benchmark datasets have been labeled by multiple (5-20) raters. Due to the high cost of manual labeling, most existing public benchmarks are in extremely small scale, containing only a few tens or hundreds of videos. This insufficient benchmark scale causes multiple issues such as unstable evaluation or high sensitivity in train-test splits. We present Mr. HiSum (https://github.com/MRHiSum/MR.HiSum), a large-scale dataset for video highlight detection and summarization, containing 31,892 videos and reliable labels aggregated over 50,000+ users per video. We empirically prove reliability of the labels as frame importance by cross-dataset transfer and user study.

## 1 Introduction

Nowadays, an enormous amount of videos are prevalent around us. With the widespread smart devices and popularity of video sharing platforms like YouTube, videos are easily created and shared at an unprecedented scale. Followed by the proliferation of video data, users tend to prefer watching short snippets of interesting content or video summaries rather than consuming the entire long video.

To meet this demand, researchers have studied automatic video highlight detection, to find the most engaging (interesting) moments from a long video. It is highly challenging to learn a general way of finding highlights from a variety of videos in the real world. With recent advances in visual content understanding [57, 62, 56, 26, 40, 41, 39] and sequential data modeling [11, 22, 65, 29], deep learning models have been widely applied to video highlight detection [71, 69, 27, 24, 61, 46, 52, 45, 7].

Like other machine learning tasks, video highlight detection also requires access to a quality dataset at scale, sufficiently covering the vast variety of samples from the underlying distribution, to apply a data-driven approach. To the best of our knowledge, a large-scale high-quality video highlight detection dataset has never been published, probably due to the high cost of labeling for each video.

More specifically, one major difficulty of creating a video highlight detection dataset comes from the innate subjectivity of this task. Highlights of a video are the *most interesting (engaging)* parts of the whole video, but the concept of the most interesting part is inherently subjective and differs across individuals. In other words, highlight scores labeled by a handful number of raters can be easily biased to their perspective. For this reason, video highlight detection benchmarks often employ multiple raters to annotate each video to reserve confidence in the quality of the labels.

---

[*]Equal contribution
[†]Corresponding author

37th Conference on Neural Information Processing Systems (NeurIPS 2023) Track on Datasets and Benchmarks.

| Dataset Name | Tasks[*] | # Videos | Total Duration | # Raters | # Categories |
|---|---|---|---|---|---|
| SumMe [23] | S | 25 | 1.1 hours | 15-18 | - |
| TVSum [59] | H, S | 50 | 3.5 hours | 20 | 10 |
| YouTube highlight [61] | H | 712 | 23.83 hours | 5 | 6 |
| MED-Summaries [55] | S | 160 | 9 hours | 2-4 | 15 |
| VSUMM [13] | S | 100 | 3.79 hours | 5 | - |
| TV episodes [73] | S | 4 | 7 hours | - | 1 |
| UT Egocentric [43] | S | 4 | 16.96 hours | - | 1 |
| UGSum52 [44] | S | 52 | 3.45 hours | 25 | - |
| VISIOCITY [34] | S | 67 | 71 hours | 13 | 5 |
| Video2GIF [24] | H, G | 84,754 | 7,379 hours | 1 | 18 |
| Mr. HiSum (Ours) | H, S | **31,892** | **1,788 hours** | **50,000+** | **3,509** |

Table 1: List of existing video highlight detection and summarization datasets. [*]H: Video highlight detection, S: Video summarization, G: Video to GIF generation.

| Seed \ Split | split1 | split2 | split3 | split4 | split5 | →Avg |
|---|---|---|---|---|---|---|
| Seed 1 | 59.8 | 55.9 | 53.8 | 49.6 | 58.2 | $\mathbf{55.5}_{\pm 4.0}$ |
| Seed 2 | 59.9 | 55.7 | 54.2 | 52.2 | 58.5 | $\mathbf{56.1}_{\pm 3.1}$ |
| Seed 3 | 58.6 | 56.6 | 54.3 | 52.7 | 60.0 | $\mathbf{56.5}_{\pm 3.0}$ |
| Seed 4 | 60.7 | 54.9 | 49.0 | 50.8 | 59.4 | $\mathbf{55.0}_{\pm 5.1}$ |
| Seed 5 | 56.8 | 54.0 | 53.5 | 53.2 | 56.0 | $\mathbf{54.7}_{\pm 1.6}$ |
| ↓Avg | **59.2** $\pm 1.5$ | **55.4** $\pm 1.0$ | **53.0** $\pm 2.3$ | **51.7** $\pm 1.5$ | **58.4** $\pm 1.5$ | 55.5 |

Table 2: 5-fold cross-validation for PGL-SUM on TVSum with 5 random seeds.

Tab. 1 lists widely-used benchmarks for video highlight detection (and summarization, which is another subjective sub-frame selecting task; see Sec. 2 for detailed discussion) requiring multiple annotators per video. Tab. 1 shows that most datasets are composed of a limited number of videos or number of raters. Considering the innate subjectivity of the tasks, it is doubtful if this small number of videos and raters is sufficient for training and evaluation.

For instance, YouTube Highlight [61] consists of only 712 videos from only 6 specific categories rated by 5 annotators, leaving the representativeness of the labels unsure. (This small scale is actually unreliable, demonstrated in Sec. 5.4.) TVSum [59], another popular benchmark for both tasks, has employed a larger number (20) of raters per video to provide more reliable labels. However, it consists of only 50 videos, probably due to its arduous labeling procedure. Under this circumstance, it is common to set aside 10 videos (20%) for testing, which is questionable if the evaluation is reliable, since this small test set obviously does not cover a wide range of topics. Tab. 2 illustrates high variance over random splits and initialization, where the test summarization F1-scores significantly vary from 49.0 to 60.7.

Moreover, it is often needed to set aside another set of videos as a validation set to tune hyperparameters, to decide when to stop training, or to select a model, further reducing the number of available videos for training. Not just the evaluation set, researchers have suffered from the lack of training videos. Since the community has no choice but to rely on these small datasets due to the difficulty of creating one at scale, the demand for a high-quality dataset for video highlight detection and summarization is high.

This paper breaks through this issue of data deficiency, in the number of both video samples and raters by introducing a large-scale dataset for video highlight detection and summarization labeled by statistics aggregated from an unprecedented number of viewers. We present the **M**ost-**r**eplayed **Hi**ghlight Detection and **Sum**marization (**Mr. HiSum**), which consists of 31,892 videos with highlight labels available on YouTube UI, aggregated from at least 50,000 watchers per video from YouTube-8M [1]. Particularly, we take the 'Most replayed' statistics as a proxy for the highlight label of each part. As illustrated in Fig. 1, the 'Most replayed' is a publicly visible distribution showing

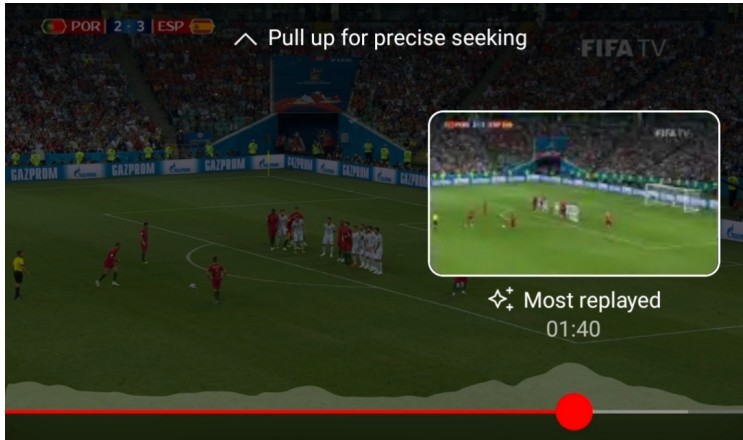

Figure 1: 'Most replayed' example

how much each part of the video has been re-watched by the users. When we consider our natural watching behavior, a primary reason for revisiting certain parts of a video is because the user thinks this part is more interesting than others. Since this distribution is collected from watching behaviors of tens of thousands of viewers, we argue that it reflects the general opinion about where people believe the most interesting or engaging, *i.e.*, highlighting parts are in each video. It is analogous to how other existing benchmark datasets have been created, where multiple (*e.g.*, 5 to 20) raters select the interesting parts of a video with their own perspectives and the highlight score is labeled as the aggregation (*e.g.*, average) of them. Based on this reasoning, we claim that the frames with high Most replayed frequency would represent the highlights of the video.

We support our claim that the Most replayed statistic is suitable to supervise highlight detection and summarization models by empirically demonstrating a superior performance on the tasks when several representative video highlight detection and summarization models are trained on the proposed dataset and transferred to existing benchmarks (Sec. 5.1).

Thanks to the sufficiently large scale and coverage over various topics, our Mr. HiSum dataset is free from the aforementioned limited reliability issue caused by extremely small validation and test sets. Using our large dataset, we empirically analyze the relationship between the reliability of evaluation and the validation and test set size. As a teaser, our study implies that at least 1,000 videos are needed to evaluate a model robustly, which is far above the number of videos in existing benchmarks.

Lastly, we examine another hypothesis that we may need a different scale of training set depending on the topics or domains of the target videos. For example, movies might require more examples than sports videos, since selecting important scenes is relatively simpler for sports, *e.g.*, making a goal for soccer. We utilize the category labels from YouTube-8M to analyze the required dataset scale to train a summarization model for each category. Our analysis indicates that video categories with relatively homogeneous formats, such as cooking videos, indeed require fewer training samples. This will become a meaningful reference for future research on category-specific video summarization.

Our main contributions are summarized as follows:

- We present Mr. HiSum, a large-scale dataset for video highlight detection and summarization, with highlight labels aggregated from the Most replayed stats in YouTube.
- We show that the proposed dataset can be used to supervise a video summarization model.
- We verify that far larger number of videos than the current benchmarks are indeed necessary to robustly evaluate highlight detection and summarization models.
- We show that the required dataset size to train a summarization model varies by the video category, providing a meaningful reference for future research.

## 2 Problem Formulation

Our Mr. HiSum dataset is mainly a video highlight detection dataset, but it can also be applied to video summarization. Both tasks share in common that they select a subset of sub-clips from a video,

but the underlying semantics of which clips to choose is quite different. We first compare the two tasks in intuitive and technical aspects. Then, we explain how our dataset can be used for both tasks.

**Video highlight detection** is a task to retrieve the most interesting and engaging parts from a whole video, while **video summarization** is a task to select sub-clips within a budget to reflect the whole synopsis or context of a long video. At a glance, these two tasks might look similar in terms of their inputs and outputs; that is, both tasks take a long video as input, and score each frame (or clip) indicating its likelihood of selection. The top-scored frames within the budget are selected and construct a set of highlights or a summary, respectively.

However, the actual meaning of the score is quite different. The highlight score indicates how much the clip is an interesting moment, while the score for summarization (also known as importance score) implies how important the frame is to reflect the entire story. These two scores are not identical, since the most interesting (engaging) parts of a video may not necessarily represent the whole video synopsis. For example, movie fans may consider a scene with a beautiful scenery as a highlight, but it may not be included in the summary if it is not highly relevant to the whole story.

Despite the clear distinction between the two tasks, however, their labels are often used interchangeably. For example, the widely used benchmark dataset, TVSum [59], provides human annotations for the summarization task. However, researchers have widely used this dataset to train and evaluate video highlight detection models [69, 27, 45, 7] as well. This implies that the positively labeled frames in both tasks may be still quite correlated, although they are not identical. In other words, frames with high summary importance may tend to be more interesting than those with lower scores, and highlight scenes tend to take part in the summary video more than others.

Most replayed statistics in our dataset reflects the nature of highlight scores, as users replay some part of a video when they are highly engaged with it. Since there is potentially some correlation between the two tasks, we pose that Mr. HiSum can also be reasonably utilized to supervise a video summarization model. We empirically support this claim by demonstrating a superior zero-shot performance when a summarization model is trained on our dataset and transferred to existing summarization benchmarks (Sec. 5.1). Since neither of these tasks had a high quality large-scale dataset, Mr. HiSum would be used for both tasks to improve performance with more reliable evaluation (Sec. 5.4).

## 3 Related Work

### 3.1 Video Datasets

**General Video Datasets.** Large-scale video datasets [36, 47] are leading the advances in video understanding [9, 68], following the path of how ImageNet [14] has contributed to image understanding [26, 62, 63, 57, 38]. For video classification or action recognition, YouTube-8M [1, 42] provides audio-visual features of 6.1M videos with labels on 3,862 classes from knowledge graph entities. Kinetics [58] contains 650,000 video clips with labels on 700 classes. Something-Something v2 [21] includes 220,847 labeled video clips of basic actions with everyday objects. Earlier datasets, like UCF101 [60], Sports-1M [33], and ActivityNet [15] provide $O(10^5)$ videos.

**Video Highlight Detection Datasets.** Due to the subjective nature of 'interest', video highlight detection needs multiple raters to cover various opinions. YouTube Highlight [61] contains 712 YouTube videos from 6 categories, with highlight labels on each segment annotated by 5 raters. Some video summarization datasets have been applied to highlight detection task as well. TVSum [59] has been used for highlight detection by converting the ground truth importance scores to highlight labels for each segment using accompanied shot information [53, 12]. Video2GIF [24] is a large-scale dataset which consists of 100k video-GIF pairs on the web. Unlike other datasets that a given video is labeled by raters, this dataset has been created in the opposite way, mapping from an existing GIF to its original source video. Unfortunately, a large portion of this dataset is no longer available, since its source website (GIFSoup) is down.

**Video Summarization Datasets.** Due to the innate subjectivity, most datasets for video summarization, listed in Tab. 1, have focused more on providing reliable annotations than increasing their volume. SumMe [23] is composed of 25 videos ranging from 1 to 6 minutes annotated by 15–18 raters. TVSum [59] contains 50 videos of 2–10 minutes collected from 10 categories, with importance scores for each equal-length shot labeled by 20 raters. In addition to these two most popular datasets, MED-

Summaries [55] provide 160 videos of 1–5 minutes from 15 categories labeled by 2–4 raters with contextual importance to their categories. VSUMM [13] contains 50 videos from YouTube and 50 videos from Open Video Project, labeled on manually selected frames by 5 raters. TV episodes [73] have 4 videos with summary labels generated from text annotations. UT Egocentric [43] contains 4 egocentric videos of 3–5 hours, annotated with descriptions of objects in the scenes. UGSum52 [44] and VISIOCITY [34] contain 52 and 67 videos labeled by multiple raters, respectively, but they are no longer publicly available at the time of this writing.

The datasets listed in Tab. 1 have critical issues. The summarization datasets such as SumMe or TVSum are absolutely small compared to other general video datasets, in spite of the innate complexity of the summarization task itself. Video highlight datasets, on the other hand, tend to have more videos, while their labels have been provided by very few raters, resulting in limited labeling consistency. Our proposed dataset, Mr. HiSum, consists of 31,892 videos with at least 50,000 views per video, overcoming the drawbacks of these existing datasets.

### 3.2 Video Highlight Detection and Summarization Methods

In spite of inherent subjectivity of discriminating relative importance among frames [74, 77], recent supervised learning approaches have shown promising performance on video highlight detection and summarization. For video highlight detection, ranking approaches [72, 31, 24, 61] learn to score highlighted segments higher than the rest. Recently, attention-based methods [67, 7, 70] are also widely studied. SL-module [70] introduces a Transformer-based model. For video summarization, RNNs [75, 76], LSTMs [74, 77, 66, 28], or attention-based models [16, 4, 19, 50, 78] have been proposed to capture temporal dynamics. VASNet [16] is the first model to apply attention to video summarization, and PGL-SUM [4] combines local and global attention. He *et al.* [25] and Narasimhan *et al.* [49] propose multi-modal approaches.

Since the data has been the bottleneck for both tasks [23, 59, 61], some recent approaches tackle this environment. Jiang *et al.* [30] utilizes additional moment localization datasets to address data scarcity. Unsupervised [3, 2, 8, 54, 69, 71, 5, 32, 48] and weakly supervised [27] methods leverage a large-scale dataset like YouTube-8M [54] or Instagram [69].

### 3.3 Category-specific Video Summarization

Category-specific video summarization often utilizes additional category information to score scenes [51]. For instance, early attempts in sports summarization have utilized sports-specific information, such as the game rule [10] and a specific action set [17, 6]. For egocentric video summarization, Lee *et al.* [43] focuses on finding important objects and people from first-person perspective. Narasimhan *et al.* [49] proposes a summarization model for instructional videos, utilizing the instruction steps from Automatic speech recognition (ASR). Fukusato *et al.* [18] evaluates anime summarization by measuring correspondence between anime and comics. Kausha *et al.* [35] learns domain-specific importance while capturing representative shots. Some summarization and highlight detection datasets provide additional category annotations [59, 61, 55, 34].

## 4 Most-replayed Highlight Detection and Summarization Dataset

We describe our proposed Mr. HiSum dataset in more detail, including how we collect and preprocess the source videos and labels, and the distribution of category annotations.

### 4.1 Most Replayed Statistics

The 'Most replayed' is a publicly visible YouTube feature for select videos, showing the frequency of "rewatched" views within the video aggregated over all watchers [20] along with a scroll bar. Fig. 1 is an example of a world cup soccer video, where the highest Most replayed scene is when the kicker is ready to free-kick at the last moment of the game, which eventually became a goal. As this example illustrates, when a certain part of a video has been replayed many times, the part is what people find interesting and engaging, which exactly fits the definition of highlight detection. Therefore, the Most replayed is a reasonable statistic to supervise a highlight detection model. This Most replayed distribution is reliable when the number of users aggregated is sufficiently large. For

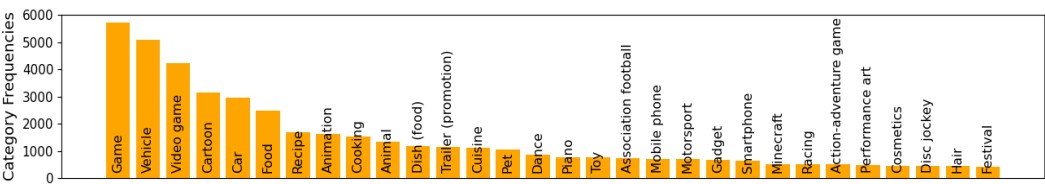

Figure 2: Frequency of the top 30 categories in Mr. HiSum.

this reason, we filter YouTube-8M videos with at least 50,000 views, so that we can safely assume that the importance of each frame is not biased towards just a few users.

## 4.2 Data Collection

Mr. HiSum utilizes a subset of the YouTube-8M dataset [1], which provides frame-level visual features at 1 fps for 6.1M videos, extracted from the Inception-v3 [63] trained on ImageNet [14] and PCA-ed to 1024D. As YouTube-8M provides the URL of each video, we crawl the metadata of the videos including the Most replayed stats, view counts, and video length. The Most replayed stats are provided as a sequence of 100 normalized scores (between 0 and 1), where each corresponds to the relative view frequency of 100 uniformly segmented clips. For instance, if a video is 600-second-long, each score indicates the relative play frequency of each 6-second-long segment. The Most replayed scores are aligned with the sequence of features at 1 fps and then used as the highlight score label.

However, we could not use all videos in YouTube-8M. Deleted videos are excluded since Most replayed and other metadata are no longer available. We also exclude videos longer than 300 seconds, because YouTube-8M provides visual features only up to 300 seconds and cropping Most replayed would damage its meaning of relative importance. Also, as the video summarization task focuses on visual cues, we exclude videos in the music category, which have similar visual cues but have significant changes in Most replayed statistics depending on the audio. As a result, 31,892 videos remain in our Mr. HiSum dataset. Their view counts are ranging from 50k to 290M. The duration of the videos is between 121 and 300 seconds, with 201.9 seconds on average.

## 4.3 Training and Inference for Video Highlight Detection and Summarization

As mentioned in Sec. 2, Mr. HiSum dataset can be used for both video highlight detection and summarization. A model is trained to estimate the Most replayed score for all frames, and they can be applied to both tasks as follows.

For video highlight detection, we first uniformly divide the input video into 5-second-long shots and aggregate frame scores within each shot by taking the average. The top $\rho \in \{15\%, 50\%\}$ shots are nominated as ground truth highlights, following previous works [27, 53, 74]. Mean Average Precision (MAP) is used to measure the performance.

For video summarization, we follow a widely-used evaluation scheme by Zhang *et al.* [74]. Specifically, predicted frame importance scores are aggregated to the shot-level using KTS [55] boundary information. Then, the top-scored shots are chosen by solving 0/1 knapsack within a given budget (*e.g.*, 15% of the original video length), where the chosen shots construct a video summary. In literature, F1 has been widely used to evaluate video summarization models.

## 4.4 Category Annotations

Mr. HiSum is collected from a subset of YouTube-8M, which provides category labels annotated from 3,862 knowledge graph entities. Videos in Mr. HiSum are annotated with 3,509 entities among them, and 153 entities have 100 or more examples. Fig. 2 demonstrates the frequency of the top 30 categories. One video may have multiple category annotations, since YouTube-8M is a multi-labeled classification dataset. In Sec. 5.5, we conduct an additional experiment to compare the difficulty level of video summarization across various domains, leveraging the vast size of our dataset.

| | TVSum | | | | YouTube highlight | | |
|---|---|---|---|---|---|---|---|
| Category | Target | Zeroshot | Fine-tuned | Category | Target | Zeroshot | Fine-tuned |
| Vehicle tire | 80.3±3.9 | 77.2±0.9 | **85.3±2.0** | dog | 48.9±3.7 | **56.1±2.1** | 50.8±3.2 |
| Vehicle unstuck | 67.1±2.5 | 64.6±1.7 | **69.3±2.8** | gymnastic | 62.5±2.4 | 58.8±1.3 | **67.2±3.0** |
| Grooming animal | 83.4±2.8 | 76.0±2.9 | **83.9±3.7** | parkour | 74.0±4.5 | 66.1±2.1 | **78.3±2.1** |
| Making sandwich | 75.2±1.5 | 63.4±0.6 | **76.6±1.4** | skating | 36.5±4.1 | **43.5±1.3** | 42.0±2.6 |
| Parkour | 61.7±1.5 | **61.8±1.1** | 61.7±0.6 | skiing | **62.7±1.4** | 56.3±0.6 | 61.8±1.0 |
| Parade | 77.5±2.6 | 70.3±0.9 | **79.0±1.5** | surfing | **74.7±2.4** | 66.9±1.6 | 71.6±0.8 |
| Flash mob | 61.5±1.7 | **69.5±0.5** | 62.5±0.7 | | | | |
| Beekeeping | **79.6±0.7** | 66.4±1.6 | 75.0±2.9 | | | | |
| Bike tricks | 67.1±4.2 | **77.4±1.9** | 73.9±2.7 | | | | |
| Dog show | 59.4±2.7 | **68.0±0.7** | 62.6±2.3 | | | | |
| Average | 71.3±1.5 | 69.5±0.4 | **73.0±0.7** | Average | 59.9±1.7 | 57.9±1.2 | **62.0±1.3** |

Table 3: Highlight detection performance of SL-module [70] in MAP, trained on the *Target*, Mr. HiSum (*Zeroshot*), and *Fine-tuned* on the target after trained on Mr. HiSum.

# 5 Experiments

We verify the validity of our Most replayed labels as a proper source to supervise video highlight detection and summarization by transferring to existing datasets (Sec. 5.1). Then, we provide benchmark scores of the representative models on our dataset (Sec. 5.3). We empirically verify an ideal scale of summarization datasets for validation and evaluation (Sec. 5.4) as well as for category-specific training (Sec. 5.5).

## 5.1 Cross-dataset Verification

**Motivation.** To demonstrate that the Most replayed stats can supervise video highlight detection and summarization models, we conduct cross-dataset evaluation experiments. Concretely, we compare the performance of several representative highlight detection and summarization models trained on our dataset *vs.* those trained on existing small benchmarks, tested on the existing ones. If the Most replayed labels provide adequate frame importance ground truth to train and validate these models, they would perform well on existing benchmarks through zero-shot or fine-tuned transfer learning.

**Experimental Settings.** We use the Inception-v3 [63] features reduced to 1024D by PCA, following YouTube-8M [1]. For a fair comparison, the same set of features are extracted for TVSum [59], SumMe [23], and YouTube highlight [61], allowing seamless transfer learning across datasets. We choose the current state-of-the-art models, PGL-SUM [4] and VASNet [16] for video summarization and SL-Module [70] for video highlight detection. We exclude videos with artificial scenes (*e.g.*, video games, cartoons, and animations) for this experiment, since the target datasets we transfer to contain only realistic videos, leaving 22,743 videos for training. For highlight detection, we set-aside 20% (TVSum) and 30% (YouTube highlight) of the videos within each category for testing, following previous works [70, 61]. For summarization (TVSum and SumMe), we use 5-fold cross-validation (20% videos used for testing at each trial) following existing works [74, 4, 16]. From the 20 labels provided in TVSum, we take the segments with top $\rho = 50\%$ average scores as ground truth following Hong *et al.* [27], and measure top-5 MAP. YouTube highlight does not need this processing, since it provides a single ground truth label (with different $\rho$) for each video. We repeat all experiments with 5 random seeds.

**Results and Discussion.** Tab. 3 shows the cross-dataset transfer performance on video highlight detection benchmarks, TVSum [59] and YouTube highlight [61]. On average, the models pretrained on Mr. HiSum then *fine-tuned* on the target perform the best, as expected. Interestingly, however, the *zeroshot* often outperforms the *fine-tuned* on some categories. This is because Mr. HiSum contains a large number of samples in those categories, *e.g.*, 'Bike tricks', 'Dog show', 'dog', or 'skating'. In this case, fine-tuning on a small target dataset unlearns some features learned from Mr. HiSum, slightly lowering the performance. The *fine-tuned* still outperforms the *target*, indicating that the learned features from pre-training still play some role after fine-tuning.

| Datasets | | Model | |
|---|---|---|---|
| Training | Test | PGL-SUM | VASNet |
| TVSum | TVSum | 55.5±0.7 | 54.2±0.9 |
| Mr. HiSum | TVSum | **57.1±0.7** | **57.1±1.0** |
| SumMe | SumMe | 41.7±3.2 | 41.9±3.3 |
| Mr. HiSum | SumMe | **42.3±2.1** | **42.6±1.3** |

Table 4: Video summarization performance on Cross-dataset transfer.

| Agreement | A | D | Count | Percentage | |
|---|---|---|---|---|---|
| Agree | 6 | 0 | 59 | 23.6% | |
| | 5 | 1 | 61 | 24.4% | **64.8%** |
| | 4 | 2 | 42 | 16.8% | |
| Neutral | 3 | 3 | 39 | 15.6% | 15.6% |
| Disagree | 2 | 4 | 25 | 10.0% | |
| | 1 | 5 | 12 | 4.8% | 19.6% |
| | 0 | 6 | 12 | 4.8% | |

Table 5: User Study Results. A:D is the number of raters whose answer is same:opposite to our label.

On categories with a relatively small number of videos in Mr. HiSum such as 'gymnastics', 'skiing', 'surfing', 'Beekeeping', or 'Making sandwich', *zeroshot* underperforms compared to *target*. Once *fine-tuned*, however, the performance gets significantly better, sometimes outperforming *target* or being comparable. This implies that pre-training on large-scale data is beneficial in most cases, unless we target a very specific topic that is not well-represented in the large dataset.

Tab. 4 compares the video summarization performance on TVSum [59] and SumMe [23], when the models are trained on our dataset *vs.* the corresponding training sets. The results demonstrate that for all models, training on our Mr. HiSum dataset exceed the ordinary F1-scores both on TVSum and SumMe.

From these experiments, we conclude that the Most replayed statistics does effectively supervise highlight detection and summarization models, indicating that Mr. HiSum provides reliable ground truth labels on these tasks.

### 5.2  User Study

**Experimental Setup.** To verify that our label is a reasonable proxy highlight score, we conduct the following user study. We select 25 videos from our dataset and choose 10 5~10-second-long clips in each video, such that 5 of them are highlights and the other 5 are not. We recruit 30 annotators in total and each of them has been assigned with 5 videos. They are instructed to watch each video fully, and then are given 10 candidate clips per video to rate. They are asked to choose 5 most interesting (engaging) ones among them. Each clip is marked either it is a highlight or not by 6 annotators. If the majority vote is the same as the Most replayed label in our dataset, the clip is scored 'Agree'. If the majority vote is the opposite, it is scored 'Disagree'. If the votes are 3:3, it is considered 'Neutral'.

**Results and Discussion.** Tab. 5 shows that the labels for 64.8% of the clips are in accordance with the annotators, while only 19.6% are disagreed, leaving 15.6% as the gray area. From the significantly larger portion of the 'Agree' than 'Disagree', we conclude that the Most replayed labels in our dataset reasonably reflect the general human perception of video highlights.

### 5.3  Benchmarks on Mr. HiSum Dataset

We provide performance metrics of several representative video highlight detection and summarization models on our dataset for reference. We report two metrics: F1 score (widely-used for summarization) and Mean Average Precision (MAP; widely-used for highlight detection). We set aside 2,000 videos for testing in this experiment.

| Model | F1 | MAP$_{\rho=50\%}$ | MAP$_{\rho=15\%}$ |
|---|---|---|---|
| PGL-SUM [4] | **55.89**±**0.04** | **61.60**±**0.14** | **27.45**±**0.15** |
| VASNet [16] | 55.26±0.05 | 58.69±0.30 | 25.28±0.40 |
| SL-module [70] | 55.31±0.09 | 58.63±0.13 | 24.95±0.13 |
| DSNet [78] | 50.78±0.16 | 57.31±0.18 | 24.35±0.34 |
| iPTNet [30] | 50.53±0.16 | 55.53±0.25 | 22.74±0.13 |

Table 6: Benchmark on Mr. HiSum dataset

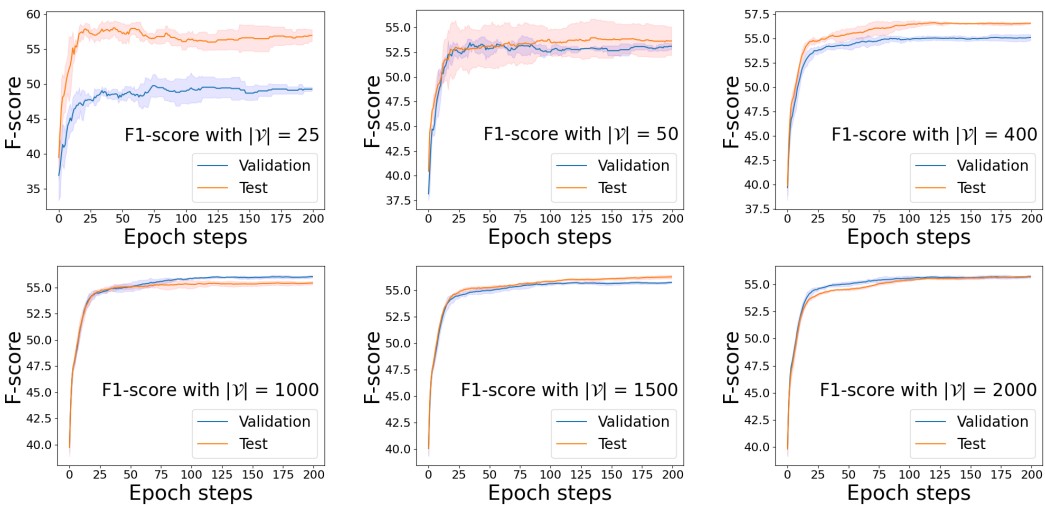

Figure 3: F1-Scores with various validation set sizes $|\mathcal{V}|$ (Shaded area indicates $\pm 1\sigma$ from the mean.)

Tab. 6 reports the performance on our test set. PGL-SUM shows the best performance on all metrics. It is noteworthy that the standard deviations on the metrics are relatively lower when evaluated on Mr. HiSum (Tab. 6), compared to the case on SumMe, TVSum or YouTube highlight reported in Tab. 3–4. This reconfirms that evaluating on a sufficient number (2,000) of videos allows a significantly more robust evaluation and avoids performance fluctuations due to random seeds.

## 5.4 Analysis on the Desired Validation and Evaluation Set Size

It is now clear that tens of videos are insufficient to cover the vast video space. Then, how many videos do we need for robust training and evaluation? We aim to answer this question as a reference for future dataset creators. We admit that it is not just a matter of quantity but diversity of those samples is also important. Since YouTube-8M, where we sample from, is a carefully designed dataset to cover wide range of 3,862 topics with at least 100 videos per category, our analysis provides a more optimistic bound for a minimum number of videos required to achieve certain level of confidence, where the video pool is relatively well-representative of the real world.

We vary the size of validation set $|\mathcal{V}| = \{25, 50, 400, 1000, 1500, 2000\}$ to see the robustness of validation performance throughout the training. We trace the validation and test F1-scores throughout training the PGL-SUM model on 27,892 videos from Mr. HiSum up to 200 epochs. Fig. 3 illustrates notably unstable validation and test scores with large fluctuation when $|\mathcal{V}|$ is small. The curves get smoother with larger $|\mathcal{V}|$.

## 5.5 Category-specific Analysis

**Motivation.** In general, videos in a particular topic tend to have their own way of storytelling or commonly-used structures. Such a particular tendency would make the summarization method more specialized to the category, and thus a category-specific summarization would be simpler than its general counterpart. We test a hypothesis that each category poses different level of complexity in summarization, requiring different scale of data samples to achieve certain level of accuracy.

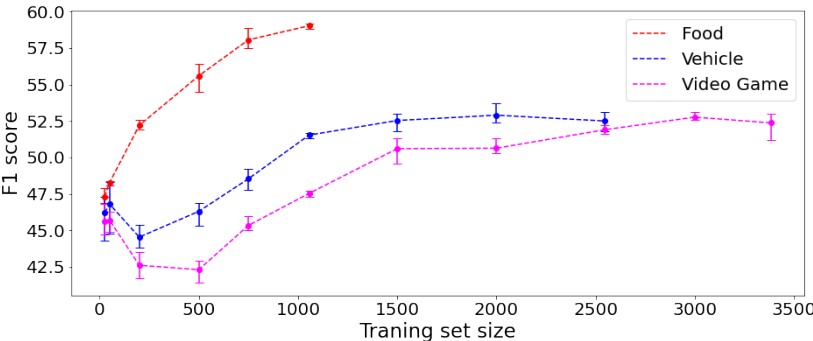

Figure 4: Summarization performance of different video categories with varied training set size

**Experimental Settings.** We focus on 3 video categories: Food, Vehicle, and Video Game. We filter videos based on class labels from our dataset, resulting in 1,456 Food videos ('food', 'cooking', and 'recipe' classes), 2,497 Vehicle videos ('vehicle' and 'car' classes), and 3,785 Video Game videos ('game' and 'videogame' classes). We use the PGL-SUM model, and set aside 200 videos for validation and test, respectively, from each category. Among the remaining videos, we vary the training set size to $\{25, 50, 400, 1000, 1500, 2000\}$ and measure F1.

**Results and Discussion.** As demonstrated in Fig. 4, the Food category shows significant performance boost with the increased training set size, while the Vehicle and Video Game categories tend to show slower improvement. Overall, Food categories require far less number of videos to achieve higher performance than the other two. We conjecture that the Food videos tend to have a similar format of demonstrating recipes step by step, and thus these videos can be summarized relatively simply. On the other hand, Vehicle and Video Game tend to have more diverse contents, making summarization harder. To conclude, the difficulty of the video summarization task varies by categories. We leave further analysis with more categories as an interesting future work.

# 6 Summary and Discussion

To address the data sparsity problem inherent in video highlight detection and summarization, we create Mr. HiSum, which consists of 31,892 videos and labels from the watching behaviors aggregated over 50,000+ viewers per video. From robustness analysis, we conclude that a validation set size of 1,000 or larger is necessary to obtain robust evaluation, disqualifying most existing datasets for these tasks. We also discover that the difficulty of the video summarization task significantly varies by video categories. With the dataset scarcity issue resolved with our dataset, we look forward to seeing further development of new models in the research community.

Our dataset also has some limitations. While the definition of replay is intuitive to understand, it is unclear how to count a replay if a segment is rewatched only partially. Unfortunately, YouTube does not provide the exact way of counting replays, so our Mr. HiSum labels may not be precise. However, we argue that this does not bring a significantly different distribution in our case, since the length of each segment is quite short. Only the adjacent segments will be affected depending on how to treat partial rewatches, while the entire distribution will remain roughly the same.

In addition, the Most replayed stats may be affected by self-fulfilling bias. That is, users might replay certain parts of the video just because the UI indicates that the parts have been rewatched by many others already. Our labels may not be completely free from such a bias. Actually, the first 50,000 viewers are not affected at all, as the Most replayed UI is shown after at least 50,000 views. After then, however, once the Most replayed distribution appears on the UI, it may be hard to assert that users were not influenced by it. Unfortunately, it is hard to measure or remove the impact of this bias, since YouTube does not provide the stats separately for the first 50,000 viewers. Considering that this is based on explicit replays, not just watching, we believe the impact of such a bias would not be significant.

## Acknowledgement

We sincerely thank Kwanseok Kim (Seoul National University) for insightful discussion. This work was supported by the New Faculty Startup Fund from Seoul National Univer- sity and by National Research Foundation (NRF) grant (No. 2021H1D3A2A03038607/50%, 2022R1C1C1010627/20%, RS- 2023-00222663/10%) and Institute of Information & communications Technology Planning & Evaluation (IITP) grant (No. 2022-0-00264/20%) funded by the government of Korea.

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
