# Supplementary Material:
# Mr. HiSum: A Large-scale Dataset for Video Highlight Detection and Summarization

**Jinhwan Sul**[*]
Georgia Institute of Technology
jsul7@gatech.edu

**Jihoon Han**[*]
Seoul National University
joon7092@snu.ac.kr

**Joonseok Lee**[†]
Seoul National University
joonseok@snu.ac.kr

## A    Appendix

The Mr. HiSum dataset is available at `https://github.com/MRHiSum/MR.HiSum`. This GitHub repository provides a Pytorch framework for video highlight detection and summarization, which will help future researchers to develop their own video summarization (or video highlight detection) model using our dataset.

### A.1    Implementation Details on Baseline Models

In the main manuscript, we evaluate five state-of-the-art video summarization and video highlight detection models on Mr. HiSum dataset. PGL-SUM [2] is trained for 200 epochs with Adam optimizer [6] using exponential learning rate decay with $\gamma = 0.97$, starting from a learning rate of $5 \times 10^{-5}$. We use $L_2$ regularizer with the weight of $5 \times 10^{-4}$ for the Adam optimizer. VASNet [4] is also trained for 200 epochs with Adam optimizer [6], and we use exponential learning rate decay with $\gamma = 0.99$ and initial learning rate of $5 \times 10^{-5}$. We use $L_2$ regularizer with the weight of $1 \times 10^{-4}$ for the Adam optimizer. SL-module [10] is trained with Stochastic Gradient Descent (SGD) for 200 epochs, and we decay learning rate once at 100 epoch from the initial learning rate 0.05 to 0.005. DSNet [12] is trained for 500 epochs with Adam optimizer with a weight decay of $1 \times 10^{-5}$ and we set the learning rate to $1 \times 10^{-6}$. iPTNet [5] is trained for 100 epochs with Adam optimizer with a weight decay of $1 \times 10^{-4}$ and we set the learning rate to $1 \times 10^{-5}$. All models are trained and evaluated on NVIDIA A6000 GPU with CUDA 11.0. The original SL-module [10] uses C3D [9] as a video clip encoder. However, as Mr. HiSum already provides image feature(ImageNet [3] feature extracted from Inception-v3 [8] network and reduced to 1024 dimension through PCA), we omit the encoder part of the model and only trained the rest of the layers of the model. Other hyperparameters and training details follow the original released configuration of each model.

### A.2    License of Assets

This dataset is licensed under Creative Commons Attribution 4.0 International (CC BY 4.0)[3], following the YouTube-8M [1] dataset. Furthermore, all the Mr. HiSum dataset users must comply with YouTube Terms of Service[4] and YouTube API Services Terms of Service[5]. Also, our Pytorch

---

[*]Equal contribution

[†]Corresponding author

[3]https://creativecommons.org/licenses/by/4.0/

[4]https://www.youtube.com/static?template=terms

[5]https://developers.google.com/youtube/terms/api-services-terms-of-service#agreement

Submitted to the 37th Conference on Neural Information Processing Systems (NeurIPS 2023) Track on Datasets and Benchmarks. Do not distribute.

video summarization framework source code has referred to the original repository of PGL-SUM[6], VASNet[7], and SL-modules[8]. Therefore, every part of the code from the original repository follows the corresponding license. We provide our code under the same custom academic and non-commercial use license from PGL-SUM.

### A.3  Author Statement

We bear all responsibility in case of violation of rights. The information provided in the paper and the supplementary material is truthful and accurate. Also, we adhere to and comply with YouTube Terms of Service and YouTube API Services Terms of Service.

### A.4  Hosting and Maintenance Plan

Mr. HiSum is hosted, managed, and maintained by the authors of the paper, Jinhwan Sul, Jihoon Han and Joonseok Lee. We host our dataset on `https://github.com/MRHiSum/MR.HiSum` as mentioned in the main manuscript, and will handle all the inconvenience through GitHub issue or email. Mr. HiSum may be updated when more videos in the YouTube-8M [1] dataset accumulate further view counts, or the YouTube-8M itself is refreshed.

## B   Datasheets for the Mr. HiSum dataset

### B.1  Motivation

**Q1: For what purpose was the dataset created?** Was there a specific task in mind? Was there a specific gap that needed to be filled? Please provide a description.

**A1**: Mr. HiSum is created to introduce a large-scale dataset that can supervise important scenes of a video using YouTube's Most Replayed Statistics. This sufficiently large dataset aims to solve the video summarization and video highlight detection task. Compared to the prior benchmarks which consist of just tens or hundreds of examples, Mr. HiSum with 30k+ videos lets machine learning models conduct a much stabler training and evaluation, with less sensitivity in train-test splits.

**Q2: Who created the dataset (e.g., which team, research group) and on behalf of which entity (e.g., company, institution, organization)?**

**A2**: Mr. HiSum released on June 7th, 2023, was created by Jinhwan Sul, Jihoon Han, and Joonseok Lee, from Graduate School of Data Science at Seoul National University.

**Q3: Who funded the creation of the dataset?** If there is an associated grant, please provide the name of the grantor and the grant name and number.

**A3**: This work was supported by National Research Foundation grants (2021H1D3A2A03038607, 2022R1C1C1010627) and Institute of Information & communications Technology Planning & Evaluation (IITP) grant (No. 2022-0-00264), funded by the Korea government.

### B.2  Composition

**Q1: What do the instances that comprise the dataset represent (e.g., documents, photos, people, countries)?** Are there multiple types of instances (e.g., movies, users, and ratings; people and interactions between them; nodes and edges)? Please provide a description.

**A1**: The instances in Mr. HiSum are Most Replayed statistics (See Sec. 4.1) of videos from YouTube-8M. Videos in Mr. HiSum are annotated with 3,509 entities such as "Game", "Vehicle", "Food", etc. Among them, 153 entities have 100 or more examples.

---

[6]PGL-SUM repository link: https://github.com/e-apostolidis/PGL-SUM
[7]VASNet repository link: https://github.com/ok1zjf/VASNet
[8]SL-modules repository link: https://github.com/ChrisAllenMing/Cross_Category_Video_Highlight

**Q2: How many instances are there in total (of each type, if appropriate)?**

**A2**: The dataset contains 31,892 videos in total, with a total duration of videos being 1,788 hours. The dataset contains 3,509 categories.

**Q3: Does the dataset contain all possible instances or is it a sample (not necessarily random) of instances from a larger set? If the dataset is a sample, then what is the larger set? Is the sample representative of the larger set (e.g., geographic coverage)?** If so, please describe how this representativeness was validated/verified. If it is not representative of the larger set, please describe why not (e.g., to cover a more diverse range of instances, because instances were withheld or unavailable)

**A3**: Mr. HiSum is a subset of the YouTube-8M dataset. The dataset consists of 31,892 videos with at least 50,000 views at the time of dataset creation (March 2023), filtered from the YouTube-8M dataset. Since the original YouTube-8M dataset has not been produced based on view counts and it was 7 years ago, this sampling might have changed overall distribution over classes or topics. Among the summarization datasets, Mr. HiSum is the largest dataset to the best of knowledge.

**Q4: What data does each instance consist of?** "Raw" data (e.g., unprocessed text or images) or features? In either case, please provide a description.

**A4**: The Most replayed stats are provided as a sequence of 100 normalized scores (between 0 and 1), where each corresponds to the relative view frequency of 100 uniformly segmented clips. The Most replayed scores are aligned with the YouTube-8M provided sequence of features at 1 fps and then used as the importance score label.

**Q5:Is there a label or target associated with each instance?** If so, please provide a description.

**A5**: Yes, our dataset consists of the YouTube's Most Replayed statistics, used as frame importance score labels.

**Q6: Is any information missing from individual instances?** If so, please provide a description, explaining why this information is missing (e.g., because it was unavailable). This does not include intentionally removed information but might include, e.g., redacted text.

**A6**: We provided all data without any omissions.

**Q7: Are relationships between individual instances made explicit (e.g., users' movie ratings, social network links)?** If so, please describe how these relationships are made explicit.

**A7**: There are no explicit relationships between individual instances.

**Q8: Are there recommended data splits (e.g., training, development/validation, testing)?** If so, please provide a description of these splits, explaining the rationale behind them.

**A8**: We randomly split 31,892 videos into 27,892, 2,000, and 2,000 for training, validation, and test, respectively. We provide this split file in our GitHub homepage and recommend others to use this split. However, other splits may be used.

**Q9: Are there any errors, sources of noise, or redundancies in the dataset?** If so, please provide a description

**A9**: Mr. HiSum is sampled from millions of videos in YouTube-8M, therefore some videos might not be useful; *e.g.*, a video with stationary visual cue or heavily fluctuating importance scores due to the background music. However, the Most replayed itself is a statistics aggregated over more than 50,000 people, and thus provides reliable meaning of generally acceptable importance between video frames.

**Q10: Is the dataset self-contained, or does it link to or otherwise rely on external resources (e.g., websites, tweets, other datasets)?** If it links to or relies on external resources, a) are there guarantees that they will exist, and remain constant, over time; b) are there official archival versions of the complete dataset (i.e., including the external resources as they existed at the time the dataset was created); c) are there any restrictions (e.g., licenses, fees) associated with any of the external resources

that might apply to a dataset consumer? Please provide descriptions of all external resources and any restrictions associated with them, as well as links or other access points, as appropriate.

**A10**: Mr. HiSum relies on the YouTube-8M dataset. The dataset has been existing since 2016, and there is no expiration date notified. Raw videos may not be accessible if they are deleted later, but their features have been consistently available, and this is same for the labels we provide. YouTube-8M has three versions (2016, 2017, and 2018), and our dataset is created based on the latest one (2018). YouTube-8M is free of charge, as far as the users agree with its license.

**Q11: Does the dataset contain data that might be considered confidential (e.g., data that is protected by legal privilege or by doctor-patient confidentiality, data that includes the content of individuals' non-public communications)?** If so, please provide a description.

**A11**: There is no confidential data in Mr. HiSum dataset. All the videos in Mr. HiSum is a subset of the YouTube-8M dataset which is publicly available under the Creative Commons Attribution 4.0 International (CC BY 4.0) license. Also, Most replayed statistics and other metadata are publicly available data that one can obtain through the YouTube website and YouTube data API.

**Q12: Does the dataset contain data that, if viewed directly, might be offensive, insulting, threatening, or might otherwise cause anxiety?** If so, please describe why.

**A12**: No, Mr. HiSum does not contain any sensitive data since it only provides Most replayed statistics and visual features from YouTube-8M. The raw videos in YouTube-8M have been already confirmed not to contain any offensive content by its creators and competition organizers.

**Q13: Does the dataset identify any subpopulations (e.g., by age, gender)?** If so, please describe how these subpopulations are identified and provide a description of their respective distributions within the dataset.

**A13**: No, Mr. HiSum does not identify any subpopulations.

**Q14: Is it possible to identify individuals (i.e., one or more natural persons), either directly or indirectly (i.e., in combination with other data) from the dataset?** If so, please describe how.

**A14**: Individuals appearing in a public video may be identifiable, using the provided video ID. The raw video itself, however, is not part of our dataset though.

**Q15: Does the dataset contain data that might be considered sensitive in anyway (e.g., data that reveals race or ethnic origins, sexual orientations, religious beliefs, political opinions or union memberships, or locations; financial or health data; biometric or genetic data; forms of government identification, such as social security numbers; criminal history)?** If so, please provide a description.

**A15**: No, Mr. HiSum is a subset of YouTube-8M dataset and it has already filtered any sensitive or offensive content through automated classifiers.

**B.3  Collection Process**

**Q1: How was the data associated with each instance acquired?** Was the data directly observable (e.g., raw text, movie ratings), reported by subjects (e.g., survey responses), or indirectly inferred/derived from other data (e.g., part-of-speech tags, model-based guesses for age or language)? If the data was reported by subjects or indirectly inferred/derived from other data, was the data validated/verified? If so, please describe how.

**A1**: YouTube-8M dataset provides URL of the contained videos. We access the Most replayed statistics for those videos through YouTube user interface. This is directly observable to YouTube users as a form of graph for select videos along with the temporal scroll bar. We provide the source code that crawls the Most replayed statistics in `https://github.com/MRHiSum/MR.HiSum`.

**Q2: What mechanisms or procedures were used to collect the data (e.g., hardware apparatuses or sensors, manual human curation, software programs, software APIs)?** How were these mechanisms or procedures validated?

**A2**: We crawl the Most replayed statistics from YouTube using our Most replayed crawler, provided at `https://github.com/MRHiSum/MR.HiSum`.

**Q3: If the dataset is a sample from a larger set, what was the sampling strategy (e.g., deterministic, probabilistic with specific sampling probabilities)?**

**A3**: We sample videos from YouTube-8M. Deleted videos are excluded since Most replayed and other metadata are no longer available. We also filter out videos with less than 50,000 view counts since we focus on collecting reliable labels. We also exclude videos longer than 300 seconds, because YouTube-8M provides visual features only up to 300 seconds and cropping Most replayed would damage its meaning of relative importance. Also, as the video summarization task focuses on visual cues, we exclude videos in the music category, which have similar visual cues but have significant changes in Most replayed statistics depending on the audio.

**Q4: Who was involved in the data collection process (e.g., students, crowdworkers, contractors) and how were they compensated (e.g., how much were crowdworkers paid)?**

**A4**: Only the three authors participated in the collection process.

**Q5: Over what timeframe was the data collected?** Does this timeframe match the creation timeframe of the data associated with the instances (e.g., recent crawl of old news articles)? If not, please describe the timeframe in which the data associated with the instances was created.

**A5**: The Most replayed statistics in Mr. HiSum dataset was collected on March 2023. Although the Most replayed statistics change over time, we confirm that the relative frame importance does not significantly change when aggregated over 50,000 samples.

**Q6: Were any ethical review processes conducted (e.g., by an institutional review board)?** If so, please provide a description of these review processes, including the outcomes, as well as a link or other access point to any supporting documentation.

**A6**: N/A (No animal subject involved.)

**Q7: Does the dataset relate to people?** If not, you may skip the remaining questions in this section.

**A7**: No.

**Q8: Did you collect the data from the individuals in question directly, or obtain it via third parties or other sources (e.g., websites)?**

**A8**: N/A

**Q9: Were the individuals in question notified about the data collection?** If so, please describe (or show with screenshots or other information) how notice was provided, and provide a link or other access point to, or otherwise reproduce, the exact language of the notification itself.

**A9**: N/A

**Q10: Did the individuals in question consent to the collection and use of their data?** If so, please describe (or show with screenshots or other information) how consent was requested and provided, and provide a link or other access point to, or otherwise reproduce, the exact language to which the individuals consented.

**A10**: N/A

**Q11: If consent was obtained, were the consenting individuals provided with a mechanism to revoke their consent in the future or for certain uses?** If so, please provide a description, as well as a link or other access point to the mechanism (if appropriate).

**A11**: N/A

**Q12: Has an analysis of the potential impact of the dataset and its use on data subjects (e.g., a data protection impact analysis) been conducted?** If so, please provide a description of this analysis, including the outcomes, as well as a link or other access point to any supporting documentation.

**A12**: N/A

## B.4   Preprocessing / Cleaning / Labeling

**Q1: Was any preprocessing/cleaning/labeling of the data done (e.g., discretization or bucketing, tokenization, part-of-speech tagging, SIFT feature extraction, removal of instances, processing of missing values)?** If so, please provide a description. If not, you may skip the remainder of the questions in this section.

**A1**: For convenience, we provide additional labels for video sumamrization. Following a widely-used evaluation scheme by Zhang et al. [11], we convert ground truth frame importance scores (Most replayed statistics in this case) into shot-level important scores using boundary information obtained by the KTS [7] algorithm. Then, the top-scored shots are chosen by solving 0/1 knapsack within a given budget (*e.g.*, 15% of the original video length), where the chosen shots construct a ground truth video summary. We provide ground truth video summary and shot boundary information as additional labels.

**Q2: Was the "raw" data saved in addition to the preprocessed/cleaned/labeled data (e.g., to support unanticipated future uses)?** If so, please provide a link or other access point to the "raw" data.

**A2**: Yes, we provide the raw data (Most replayed stats) and additional labels in `https://github.com/MRHiSum/MR.HiSum`.

**Q3: Is the software used to preprocess/clean/label the instances available?** If so, please provide a link or other access point.

**A3**: Yes, the code for KTS [7] is publicly available at https://github.com/TatsuyaShirakawa/KTS and 0/1 knapsack algorithm code we use is available at `https://github.com/MRHiSum/MR.HiSum`.

## B.5   Uses

**Q1: Has the dataset been used for any tasks already?** If so, please provide a description.

**A1**: Besides from our paper, Mr. HiSum dataset has not been used yet.

**Q2: Is there a repository that links to any or all papers or systems that use the dataset?** If so, please provide a link or other access point.

**A2**: Yes, our Mr. HiSum dataset repository, `https://github.com/MRHiSum/MR.HiSum`, presents baseline models that can be applied to the Mr. HiSum dataset.

**Q3: What (other) tasks could the dataset be used for?**

**A3**: Besides video summarization and video highlight detection, this dataset can be generally used for tasks that aim to learn relative importance between video segments. More generally, it might be useful for general video representation learning as well.

**Q4: Is there anything about the composition of the dataset or the way it was collected and preprocessed/cleaned/labeled that might impact future uses?** For example, is there anything that a future user might need to know to avoid uses that could result in unfair treatment of individuals or groups (e.g., stereotyping, quality of service issues) or other undesirable harms (e.g., financial harms, legal risks) If so, please provide a description. Is there anything a future user could do to mitigate these undesirable harms?

**A4**: This dataset is collected from and relies on YouTube-8M dataset and YouTube platform. Therefore, all the Mr. HiSum dataset users must not violate any rights stated outside of YouTube-8M CC BY 4.0 license, YouTube Terms of Service, and YouTube data API Terms of Service. As mentioned above, raw videos may not be accessible if they are deleted later.

**Q5: Are there tasks for which the dataset should not be used?** If so, please provide a description.

**A5**: No.

### B.6 Distribution

**Q1: Will the dataset be distributed to third parties outside of the entity (e.g., company, institution, organization) on behalf of which the dataset was created?** If so, please provide a description.

**A1**: The dataset is distributed through our website, `https://github.com/MRHiSum/MR.HiSum`, and it is publicly available.

**Q2: How will the dataset will be distributed (e.g., tarball on website, API, GitHub)?** Does the dataset have a digital object identifier (DOI)?

**A2**: The dataset is distributed through GitHub repository, `https://github.com/MRHiSum/MR.HiSum`.

**Q3: When will the dataset be distributed?**

**A3**: The dataset has been available since June 7, 2023.

**Q4: Will the dataset be distributed under a copyright or other intellectual property (IP) license, and/or under applicable terms of use (ToU)?** If so, please describe this license and/or ToU, and provide a link or other access point to, or otherwise reproduce, any relevant licensing terms or ToU, as well as any fees associated with these restrictions.

**A4**: The dataset is released under Creative Commons Attribution 4.0 International (CC BY 4.0) license following the YouTube-8M. Information about CC BY 4.0 license can be found in https://creativecommons.org/licenses/by/4.0/. CC BY 4.0 license allows users to copy, redistribute, remix, transform, and build upon the material for any purpose. Also, users should give appropriate credit to the Mr. HiSum dataset, should indicate if changes were made, and should not apply additional restrictions both legally and technologically. Mr. HiSum is free of charge as long as users follow this license. Furthermore, all Mr. HiSum dataset users must comply with the YouTube Terms of Service (https://www.youtube.com/static?template=terms) and YouTube API Services Terms of Service (https://developers.google.com/youtube/terms/api-services-terms-of-service#agreement).

**Q5: Have any third parties imposed IP-based or other restrictions on the data associated with the instances?** If so, please describe these restrictions, and provide a link or other access point to, or otherwise reproduce, any relevant licensing terms, as well as any fees associated with these restrictions.

**A5**: No, the dataset is licensed under Creative Commons Attribution 4.0 International (CC BY 4.0) license following the YouTube-8M dataset.

**Q6: Do any export controls or other regulatory restrictions apply to the dataset or to individual instances?** If so, please describe these restrictions, and provide a link or other access point to, or otherwise reproduce, any supporting documentation.

**A6**: No.

### B.7 Maintenance

**Q1: Who will be supporting/hosting/maintaining the dataset?**

**A1**: Mr. HiSum dataset is hosted on the GitHub repository (`https://github.com/MRHiSum/MR.HiSum`) and will be supported and maintained by the authors.

**Q2: How can the owner/curator/manager of the dataset be contacted (e.g., email address)?**

**A2**: The manager of the dataset can be contacted via email, `jsul7@gatech.edu`, `{joon7092, joonseok}@snu.ac.kr`.

**Q3: Is there an erratum?** If so, please provide a link or other access point.

**A3**: No.

**Q4: Will the dataset be updated (e.g., to correct labeling errors, add new instances, delete instances)?** If so, please describe how often, by whom, and how updates will be communicated to users (e.g., mailing list, GitHub)?

**A4**: Mr. HiSum dataset will be constantly updated by authors whenever an issue is reported. We will communicate to users via email and GitHub issue.

**Q5: If the dataset relates to people, are there applicable limits on the retention of the data associated with the instances (e.g., were individuals in question told that their data would be retained for a fixed period of time and then deleted)?** If so, please describe these limits and explain how they will be enforced.

**A5**: N/A

**Q6: Will older versions of the dataset continue to be supported/hosted/maintained?** If so, please describe how. If not, please describe how its obsolescence will be communicated to users.

**A6**: Yes, every version of the dataset will be supported via GitHub repository `https://github.com/MRHiSum/MR.HiSum`.

**Q7: If others want to extend/augment/build on/contribute to the dataset, is there a mechanism for them to do so?** If so, please provide a description. Will these contributions be validated/verified? If so, please describe how. If not, why not? Is there a process for communicating/distributing these contributions to other users? If so, please provide a description.

**A7**: Others can extend/augment/build on/contribute to the dataset by making a pull request to `https://github.com/MRHiSum/MR.HiSum`. The contributions will be validated/verified through GitHub commit history. However, contributions must adhere to and comply with CC BY 4.0 license, YouTube Terms of Service, and YouTube data API Terms of Service.