# OpenReview forum: "Mr. HiSum: A Large-scale Dataset for Video Highlight Detection and Summarization"
_NeurIPS.cc/2023/Track/Datasets_and_Benchmarks — NeurIPS 2023 Datasets and Benchmarks Poster_

### Official Review · Reviewer_2KA9 · 2023-07-02

**Rating:** 5
**Confidence:** 5
**Correctness:** The data annotation process seems a l…
**Clarity:** The paper is well-written and easy to…

**Strengths:**

(1) The proposed dataset for video summarization and video highlight detection.

(2) Compared with existing datasets, the proposed dataset contains more videos and more raters.

**Additional Feedback:**

Please see the comments above.

**Documentation:**

Code is provided in a public GitHub repo.

**Limitations:**

(1) There seems to be some ambiguous discussion about the annotation process.

(2) The baseline benchmark for the new dataset seems weak.

**Opportunities For Improvement:**

(1) There seems to be no introduction about how the annotations by 50,000+ users per video were collected, which is a very big contribution claim of the paper and the dataset.

(2) How did the authors ensure the quality of the annotations? Even if many raters have been used in building the dataset.

(3) There seem to be no instructions on how the annotation process works.

(4) There seems to be no discussion about the human cost and recruiting process.

(5) There seems to be no datasheet to discuss the proposed dataset.

(6) The newly proposed dataset heavily depends on Youtube-8M, which already made categorization, so the contribution of the proposed dataset in this categorization direction is not significant.

(7) There seem to be only three methods, which may not be enough to be considered as a benchmark on the new dataset. The video summarization task has been studied well in the past few years, and many previous works can be used as baselines:

[1] Joint video summarization and moment localization by cross-task sample transfer. CVPR 2022
[2] CLIP-It! Language-Guided Video Summarization, NeurIPS 2021
[3] DSNet: A Flexible Detect-to-Summarize Network for Video Summarization, TIP 2020
[4] Discriminative Feature Learning for Unsupervised Video Summarization, AAAI 2019
[5] Video Summarization Using Deep Neural Networks: A Survey

**Relation To Prior Work:**

The literature review seems comprehensive.

**Summary And Contributions:**

This paper mainly proposed a large-scale video summarization dataset, which contains 31,892 videos and reliable summarization labels aggregated over 50,000+ users per video.

---

> ### Author Response · Authors · 2023-08-24
> **Response to Reviewer 2KA9 (Part 1 of 2)**
>
> We sincerely thank the reviewer for their insightful comments and constructive feedback. Reflecting the reviewer’s comments, we revised the manuscript accordingly (marked _blue_ in the pdf).
>
> Before we discuss each question individually, we would like to correct the misunderstanding by the reviewer about how we collected the labels. We did _NOT_ hire actual raters, but crawled the aggregated statistics provided on the YouTube UI. Thus, we do not give any specific instructions to users. They just watch videos as they want, and their behaviors are recorded and reflected in these aggregated statistics (our labels). This would roughly resolve Q1-4, but we provide more detailed answers for each question below.
>
> ---
>
> __Q1. There seems to be no introduction about how the annotations by 50,000+ users per video were collected, which is a very big contribution claim of the paper and the dataset.__
>
> __A__. In Sec. 4.2 (Data Collection), we described how the annotations were collected. We collected YouTube’s ‘Most replayed’ statistics which is segment-wise re-watching distribution aggregated over 50,000+ users. Specifically, when a user watches a segment (1/100 uniformly split from the entire video) and rewinds again to rewatch it, this is counted as a rewatch. YouTube provides such a distribution on qualified videos on the public UI, and we crawled it. Please refer to Sec. 4 for more details.
>
> ---
>
> __Q2. How did the authors ensure the quality of the annotations? Even if many raters have been used in building the dataset.__
>
> __A__. The quality of annotations for subjective tasks like video highlight detection and summarization tasks is guaranteed by the number of users participating in the annotations, since each individual may have different opinions about what should be a highlight or what is a good summary. To reflect general ideas from various people, it is important to have a large enough sample size. 50,000 is an unprecedented scale in this line of research, as Table 1 indicates. (All other datasets had at most 25 raters per video.) For this reason, our highlight labels would be more trustworthy than any other existing benchmarks.
>
> ---
>
> __Q3. There seem to be no instructions on how the annotation process works.__
>
> __A__. As we mentioned at the beginning, we have not explicitly hired annotators, and thus we have not given any instructions to the users. This dataset is created by crawling the distribution of segment-wise rewatches shown in the YouTube UI, not by recruiting human annotators and by asking them to answer specific questions. Please refer to Sec. 4.2 for more details.
>
> ---
>
> __Q4. There seems to be no discussion about the human cost and recruiting process.__
>
> __A__. Similarly to the question above, this question is not applicable to ours, since we have not recruited any human subject for this research. Thus, there is no cost and no recruiting process to be reported.
>
> ---
>
> __Q5. There seems to be no datasheet to discuss the proposed dataset.__
>
> __A__. The datasheet is in Appendix B, in the Supplementary Material.
>
> ---
>
> __Q6. The newly proposed dataset heavily depends on Youtube-8M, which already made categorization, so the contribution of the proposed dataset in this categorization direction is not significant.__
>
> __A__. YouTube 8M is a dataset for video classification, so the categorization in the dataset is basically the class labels, mainly about video contents (topicality). Our dataset aims to video highlight detection and summarization, which need a completely different type of labels: whether each frame (or clip) is a highlight (or included in the summary). The original YouTube 8M used the categorization to balance the number of videos per class and to analyze “classification” performance depending on each category. Ours, on the other hand, analyzes the characteristics of each category when it comes to “video highlight detection” and “summarization”, which are significantly different.

---

> ### Author Response · Authors · 2023-08-24
> **Response to Reviewer 2KA9 (Part 2 of 2)**
>
> __Q7. There seem to be only three methods, which may not be enough to be considered as a benchmark on the new dataset. The video summarization task has been studied well in the past few years, and many previous works can be used as baselines.__
>
> __A__. We appreciate the reviewer for providing these relevant works. Due to the limited time for the rebuttal period, out of the four different methods that the reviewer suggested (except for the survey paper [E]), we added DSNet [C] and iPTNet [A] in Table 6 for our baseline benchmarks. (Note that iPTNet [A] takes too much time for training, so we only managed to train a model for once with a smaller number of epochs than other models. We will run additional experiments to provide the final result with standard deviation in the camera-ready version.) We excluded CLIP-It! [B] because it uses a text query as input which is not applicable to our dataset. We deprioritized CSNet [D] as it is reported underperforming than DSNet[C] and iPTNet[A], but we will consider adding this in camera-ready.
>
> [A] Joint video summarization and moment localization by cross-task sample transfer. CVPR 2022 \
> [B] CLIP-It! Language-Guided Video Summarization, NeurIPS 2021 \
> [C] DSNet: A Flexible Detect-to-Summarize Network for Video Summarization, TIP 2020 \
> [D] Discriminative Feature Learning for Unsupervised Video Summarization, AAAI 2019 \
> [E] Video Summarization Using Deep Neural Networks: A Survey

---

> > ### Comment · Reviewer_2KA9 · 2023-08-30
> > **Thank you for the response**
> >
> > I appreciate the authors for the response.
> >
> > - I apologize for my misunderstanding of the data collection process and appreciate the clarification provided. Given that the raters are not directly employed by the authors, I suggest revising the wording to something like "We extracted the aggregated statistics available through the YouTube UI," aligning with the authors' response in the rebuttal.
> > - Considering that the labels were gathered automatically online, was any human assessment conducted to gauge label quality? The current approach appears to heavily rely on the automated collection with a large dataset, yet human evaluation could offer valuable insights into the quality of the acquired data.
> > - Thank you for sharing the additional baseline results. While I understand the constraints of time, I recommend running these baselines on the entire dataset to enhance the comprehensiveness of the paper.

---

> > > ### Author Response · Authors · 2023-08-30
> > > **Response to Reviewer 2KA9's comment**
> > >
> > > __Q1. Revising the wording to clarify data collection process.__
> > >
> > > __A__. Thank you for the suggestive comments. Although this is already mentioned in Section 4.2 Data collection, line 224-225, we revised two related sentences to reduce confusion. Please check the Introduction section, line 68-69.
> > >
> > > __Q2.  Was any human assessment conducted to gauge label quality?.__
> > >
> > > __A__. We agree with the reviewer (and another reviewer, CELp, who suggested similar comments) and think that human assessment is a great idea. However, due to the short duration of the rebuttal period, we could not conduct a user study within this period. We are currently designing a human assessment study and will conduct it once IRB approves it. We will add this in camera-ready.
> > >
> > > __Q3. Running baselines on the entire dataset to enhance the comprehensiveness of the paper.__
> > >
> > > __A__. Thank you for your understanding of our time constraints. We will add results of baselines trained on the entire dataset in the camera-ready.

---

### Official Review · Reviewer_bGs5 · 2023-07-21
**Mr. Sum: Large-scale Video Summarization Dataset and Benchmark review**

**Rating:** 6
**Confidence:** 5

**Strengths:**

First and foremost, the value of this dataset is in the scale of the labels and data.  The authors are quite correct that previous datasets and small and evaluation is high-variance.

The annotation labels reflect actual human viewing behavior at scale.  The value of these viewing statistics has been well-known to industry practitioners for some time but I believe this is the first time this data has been made publicly available at usable scale.

Authors call out the class-specific nature of the summarization task with a simple experiment showing how summarization performance for different categories of videos scale with respect to the amount of training data.  Some types of video are clearly more challenging than others, and this suggests future investigation.

In table 4, the best results are generally obtained by pretraining on Mr. Sum and then using either the zero-shot or fine-tuned protocols, implying Mr. Sum pretraining is valuable relative to directly training on the target.  The fine-tuning results are generally no worse than training on target, and often significantly better.

**Additional Feedback:**

I think this dataset is quite valuable.  I'm troubled though by the framing of the problem ("summarization == highlights") and what I view as an oversimplification of the summarization problem in section 2.  I am open to being convinced, or an alternate framing which might include discussion of the limitations of current summarization benchmarks.

**Clarity:**

The paper is consistently well-written, both in the language used and the overall structure.  It is very easy to read.

**Correctness:**

The construction is simple, and the experiments are largely convincing.  Any questions or concerns are noted above.

**Documentation:**

The collection methodology is fairly clear, and the videos themselves are sourced from an existing dataset with a long OSS history (YouTube-8M).  The GitHub is linked and appears to provide code and pretrained models to reproduce the results.  I did not try the code.

**Ethics:**

I see no new concerns here.  All videos are sourced from YouTube-8M, an existing OSS dataset.  While scraping YouTube video is increasingly scrutinized for many good reasons, the authors have not included any videos beyond what was already released (by Google themselves) and no additional risk is assumed here.

**Limitations:**

The paper does not comment on societal impact, but for this particular generic task I see few issues.

**Opportunities For Improvement:**

Section 2, "Problem Formulation" is problematic.  The authors use an ill-defined notion of "importance" to define both the summarization and the highlight-detection problem.  The authors recognize that "summarization aims to generate a complete short video synopsis... OTOH, video highlight detection seeks the importance score of each video segment...", but dismiss this as a subtle point.  However, the distinction between detecting an engaging moment (a highlight) and extracting a short narrative summary is not subtle; it is quite significant.  For example, a movie trailer is explicitly designed to be highly engaging while omitting the most significant plot points.  The section concludes stating "high importance score is... highly transferable [across tasks]".  This may indeed be true, at least as shown on current benchmarks, but transferability is less than the claimed near-equivalence.

It seems more correct, for example, to argue that this dataset targets highlight detection models, where "highlight" could be defined by probability of engagement.  The 'most replayed' statistic is a good statistic for exactly this.  Then argue that empirically, this also happens to provide a strong signal for summarization.  Summarization is a much more ill-defined problem, since at the very least the correct level of detail for a summary is situation-dependent, and as the authors themselves allude to in 3.3 and 5.4, the important elements of a summary may be very domain-specific.

Epoch-to-epoch variance is a reasonable proxy statistic to use to choose eval set size, but ultimately one really cares about the variance of models trained to convergence.  If this is prohibitive, mention that you are using a single run as a proxy.

The "most replayed" data probably has a self-fulfilling bias; this should be mentioned.

While Mr. Sum finds meaningful differences between representative methods in section 5.2, there is no discussion of how to interpret these or what we might learn vs. other benchmarks.  In particular, can we compare results on the proposed benchmark with previous benchmarks?  What do we learn from these results?

In table 4, sometimes the zero-shot results are significantly better than either the target or fine-tuned results (in these cases, fine-tuning often reverts back to similar results as 'target').  Does this correlate to any obvious statistics like duration of video per-category?

**Relation To Prior Work:**

Yes, good discussion of the limits of relevant prior work is included.  The difference with prior work is very clear.  Using these crowdsourced annotations directly addresses two major concerns:  the number of raters per video needs to be large, and the number of videos needs to be large.  This is essentially impossible to achieve at scale with any other methodology.

**Summary And Contributions:**

Motivated by the lack of large-scale data for generic video summarization, the authors collect a large-scale dataset by scraping YouTube "Most Replayed" labels as a proxy signal for the importance of a video subclip.  These labels are large-scale in that they represent the viewing behavior of at least 50,000 users who have watched each video, and the dataset is large-scale in its size (orders of magnitude larger than most previous related work).  The dataset both serves as a benchmark itself, while also providing a large-scale dataset for pretraining or zero-shot inference on OOD data.  Results are presented for both video highlight detection and video summarization in several settings (pretraining, zero-shot, etc) on standard small-scale benchmarks, as well as on the Mr. Sum benchmark itself.

---

> ### Author Response · Authors · 2023-08-24
> **Response to Reviewer bGs5 (Part 1 of 2)**
>
> We sincerely thank the reviewer for their insightful comments and constructive feedback. Reflecting the reviewer’s comments, we revised the manuscript accordingly (marked _blue_ in the pdf). Also, we answer each question below. Please feel free to reply to us if you have further questions.
>
> ---
>
> __Q1. Section 2, "Problem Formulation" is problematic. (omitted)__
>
> __A__. We agree with the reviewer (and another reviewer g5Rw with a similar comment) that our dataset is more suitable to solve the highlight detection than summarization. Reflecting this point, we changed our dataset name to Mr. Highlight (from Mr. Sum) and made corresponding changes. Specifically, Sec. 2 has been revised most significantly, focusing on the distinction between the two tasks instead of their common aspects. Sec. 1 also has been revised to reflect this change of view. We really appreciate the reviewer for providing this constructive feedback; we believe this makes our paper more intuitive and strong.
>
> ---
>
> __Q2. Epoch-to-epoch variance is a reasonable proxy statistic to use to choose eval set size, but ultimately one really cares about the variance of models trained to convergence. If this is prohibitive, mention that you are using a single run as a proxy.__
>
> __A__. Reflecting the reviewer’s point, we modified Fig. 3 in Sec. 5.3 to additionally show the variance of models throughout training, indicating the standard deviation ($\pm 1 \sigma$ from the mean) across the five models we trained. Clearly, the plot shows that the variance gets smaller as the validation and test set size increases.
>
> ---
>
> __Q3. The "most replayed" data probably has a self-fulfilling bias; this should be mentioned.__
>
> __A__. We added a new Sec. 6 to describe limitations of our dataset, discussing the potential effects of self-fulfilling bias. We admit that our labels are not completely free from this bias. However, it may not be critical, since at least the first 50k viewers are not affected at all, recalling that the Most replayed feature is shown after at least 50k viewers have watched.
>
> ---
>
> __Q4. While Mr. Sum finds meaningful differences between representative methods in section 5.2, there is no discussion of how to interpret these or what we might learn vs. other benchmarks. In particular, can we compare results on the proposed benchmark with previous benchmarks? What do we learn from these results?__
>
> __A__. The main difference between our dataset and existing benchmarks is the robustness derived from the scale. With hundreds of times larger number of samples, our benchmark provides much stable evaluation. Specifically, we tried to reproduce and compare the performance of two representative methods, PGL-SUM and VASNet. As we demonstrated in Tab. 2, PGL-SUM shows high variance (min: 49.0, max: 60.7) over random splits and initialization on TVSum. A similar trend is shown for VASNet in the table below (min: 47.9, max: 61.0). Under this circumstance, it is almost impossible to tell which one is better than the other when evaluated on existing benchmarks. On our dataset, however, Tab. 6 indicates that the evaluation is significantly more robust with much smaller standard deviation. As a result, we can confirm that PGL-SUM outperforms VASNet _statistically significantly_. To sum up, our dataset enables far more robust comparison between various methods, which have not been possible with existing benchmarks.
>
> |        | split1 | split2 | split3 | split4 | split5 |
> |--------|--------|--------|--------|--------|--------|
> | Seed 1 | 58.6   | 47.9   | 52.0   | 55.5   | 53.3   |
> | Seed 2 | 57.9   | 51.2   | 54.0   | 58.0   | 56.0   |
> | Seed 3 | 53.5   | 53.4   | 57.4   | 57.9   | 51.0   |
> | Seed 4 | 61.0   | 48.5   | 52.2   | 59.0   | 52.3   |
> | Seed 5 | 57.0   | 53.0   | 50.0   | 55.0   | 50.5   |

---

> ### Author Response · Authors · 2023-08-24
> **Response to Reviewer bGs5 (Part 2 of 2)**
>
> __Q5. In table 4, sometimes the zero-shot results are significantly better than either the target or fine-tuned results (in these cases, fine-tuning often reverts back to similar results as 'target'). Does this correlate to any obvious statistics like duration of video per-category?__
>
> __A__. We think this is related to the relative number of videos in Mr. Highlight that are in the same or relevant categories of the ‘target’ datasets. For example, target categories with a relatively large number of videos in Mr.Highlight like ‘Bike tricks’, ‘Dog show’, ‘dog’, or ‘skating’ show best performance on Zeroshot. In this case, fine-tuning on a small target dataset unlearns some features learned from Mr.Highlight, slightly lowering the performance. It is noteworthy, however, that the fine-tuned performance is still higher than the Target, where no large-scale pre-training has been performed. This indicates that the learned features from pre-training on the large dataset still play some role after the fine-tuning.
>
> On the other hand, on categories with a relatively small number of videos in Mr.Highlight such as ‘gymnastics’, ‘skiing’, ‘surfing’, ‘Beekeeping’, or ‘Making sandwich’, Zeroshot underperforms compared to Target. Once fine-tuned, however, the performance gets significantly better, sometimes outperforming Target or being comparable. This implies that pre-training on large-scale data like ours is beneficial in most cases, unless we target some specific topics that are not well-represented in the large dataset.
>
> ---
>
> __Q6. The paper does not comment on societal impact, but for this particular generic task I see few issues.__
>
> __A__. We believe there are little societal concerns with our work. As our ethics reviewer (8VNJ) mentioned, Mr. Highlight uses “widely-available public videos and metrics from YouTube in non-sensitive categories”. Additionally, “the replay information does not raise privacy concerns as it is an aggregate statistics and not attributed to any particular viewers.” From the aggregated stats over 50,000+ viewers, it is not possible to discover viewing patterns of any particular individual.

---

> > ### Comment · Reviewer_bGs5 · 2023-08-30
> > **Largely addresses my concerns**
> >
> > I was already supportive of the dataset; the rebuttal (supporting analysis, and updated focus on video highlights as the primary task) I can support an accept.  I would raise my rating.

---

### Official Review · Reviewer_g5Rw · 2023-07-21
**Mr. Sum: Large-scale Video Summarization Dataset and Benchmark**

**Rating:** 6
**Confidence:** 3

**Strengths:**

This paper introduces a significantly larger dataset compared to the state-of-the-art (SOTA) dataset, which proves highly beneficial for ML training and testing. The utilization of natural watching behavior data from YouTube is an interesting approach and adds value to the research. The authors have made efforts to empirically validate the effectiveness of the labels used. Overall, the paper's findings and the dataset's scale offer promising opportunities for advancing video summarization techniques and enhancing the overall performance of machine learning algorithms in this domain. The utilization of real-world data, such as natural watching behavior from YouTube, strengthens the applicability of the proposed approach and encourages further research in this area.


**Additional Feedback:**

n/a

**Clarity:**

The paper is well-written and effectively presents its two main contributions. However, regarding the third contribution, which involves category-specific analysis, it is not entirely clear why the specific categories of food, vehicles, and video games were chosen for the analysis. In the primary experiments, artificial scenes such as video games were excluded, and examples from sports and movies were provided. Yet, the category-specific analysis focuses on the video games category without a clear rationale. By providing a clear explanation for the selection of specific categories and their relevance to the research objectives, the paper can offer a more coherent and well-supported analysis of category-specific implications.


**Correctness:**

Dataset is constructed in a sound way by utilizing publicly available YouTube replay statistics.

**Documentation:**

Dataset includes documentation and intended uses; a URL for reviewer access to the dataset.

**Ethics:**

This large dataset has the potential to make a significant impact on state-of-the-art (SOTA) machine learning research. Consequently, it is crucial to ensure that the dataset is unbiased and accurately represents the video summaries in a fair manner. At present, the dataset solely relies on YouTube replay statistics without undergoing review or verification to assess its fairness or potential biases. Thorough checks may identify any potential sources of bias, such as demographic or geographic biases, in the collected statistics.


**Limitations:**

This large dataset has the potential to make a significant impact on state-of-the-art (SOTA) machine learning research. Consequently, it is crucial to ensure that the dataset is unbiased and accurately represents the video summaries in a fair manner. At present, the dataset solely relies on YouTube replay statistics without undergoing review or verification to assess its fairness or potential biases. Thorough checks may identify any potential sources of bias, such as demographic or geographic biases, in the collected statistics.


**Opportunities For Improvement:**

It is an intriguing concept to leverage natural watching behavior as labels for Video summarization. However, it may be more suitable for video highlight detection rather than comprehensive video summarization. A video summary is expected to convey the entire story of the video, whereas the 'most replayed statistics' tend to capture highlights effectively but may lack comprehensiveness.
The paper lacks a detailed discussion of the statistics related to YouTube's natural watching behavior, which is crucial to consider. To ensure the quality of the labels, it is important to analyze the percentage of complete views vs. partial views out of the 50,000+ views per video. Many views might be partial ones, where viewers only watch the initial part before leaving. To serve as an effective video summarization method, the selected frames or video snippets should provide a cohesive and coherent narrative of the entire video. While leveraging natural watching behavior is a promising approach, the proposed method may work better for identifying highlights within a video rather than generating comprehensive summaries. A more in-depth examination of these issues would significantly enhance the dataset's utility.
Additionally, assurance or validation is needed to avoid potential biases or (un)fairness of the video summarization dataset when using natural watching behavior as a basis for video summarization. By addressing these concerns and providing more detailed insights, the proposed approach could be refined and more suitable for a broader range of applications.


**Relation To Prior Work:**

Yes.

**Summary And Contributions:**

This paper introduces a large-scale video summarization dataset called "Mr. Sum," comprising 31,892 videos with labels derived from the natural watching behaviors of over 50,000 viewers per video. The motivation behind creating this dataset is to address the scarcity of large-scale video summarization datasets, as current state-of-the-art datasets often suffer from being relatively small.
The labels in the Mr. Sum dataset are collected based on users' genuine watch behaviors, particularly focused on segments of interest that users tend to replay. This approach contrasts with datasets relying on annotations from paid annotators. The authors conducted empirical experiments to demonstrate that these natural watch behavior labels are effective for supervising video summarization models.
Additionally, the paper highlights the significance of considering video category information during summarization model training. Understanding the influence of different video categories can impact the performance.

---

> ### Author Response · Authors · 2023-08-24
> **Response to Reviewer g5Rw (Part 1 of 2)**
>
> We sincerely thank the reviewer for their insightful comments and suggestions. Reflecting the reviewer’s comments, we revised the manuscript accordingly (marked _blue_ in the pdf). Also, we answer each question below. Please feel free to reply to us if you have further questions.
>
> ---
>
> __Q1.It is an intriguing concept to leverage natural watching behavior as labels for Video summarization. However, it may be more suitable for video highlight detection rather than comprehensive video summarization. A video summary is expected to convey the entire story of the video, whereas the 'most replayed statistics' tend to capture highlights effectively but may lack comprehensiveness. (omitted for Q2) To serve as an effective video summarization method, the selected frames or video snippets should provide a cohesive and coherent narrative of the entire video. While leveraging natural watching behavior is a promising approach, the proposed method may work better for identifying highlights within a video rather than generating comprehensive summaries. A more in-depth examination of these issues would significantly enhance the dataset's utility.__
>
> __A__. We agree with the reviewer (and another reviewer bGs5 with a similar comment) that our dataset is more suitable to solve the highlight detection than summarization. Reflecting this point, we changed our dataset name to Mr. Highlight (from Mr. Sum) and made corresponding changes. Specifically, Sec. 2 has been revised most significantly, focusing on the distinction between the two tasks instead of their common aspects. Sec. 1 also has been revised to reflect this change of view. We really appreciate the reviewer for providing this constructive feedback; we believe this makes our paper more intuitive and strong.
>
> ---
>
> __Q2. The paper lacks a detailed discussion of the statistics related to YouTube's natural watching behavior, which is crucial to consider. To ensure the quality of the labels, it is important to analyze the percentage of complete views vs. partial views out of the 50,000+ views per video. Many views might be partial ones, where viewers only watch the initial part before leaving.__
>
> __A__. Most replayed stats count the number of “re”plays, not just watches, of a particular time window within each video, where the length of this time window is 1/100 of the video length. That is, each segment is counted as replayed only when a user watches some part once and intentionally rewinds back to rewatch the same part again. We revised Sec. 4.1 to better reflect the concept of rewatch.
>
> As the reviewer mentioned, it is possible that the user has rewatched some segment only partially. YouTube has not released how they count such a partial rewatch. In our case, since each segment is short enough (up to 3 seconds, since all videos we deal with are at most 5-minute-long; see Sec. 4.2), this does not really matter practically. Only the adjacent segments will be affected depending on how to treat partial rewatches, while the entire distribution will remain roughly the same. For the reviewer’s example, a viewer who watches only the initial part and leaves will not be counted if she does not rewind at all. If she replays some parts, the corresponding segments will be counted.

---

> ### Author Response · Authors · 2023-08-24
> **Response to Reviewer g5Rw (Part 2 of 2)**
>
> __Q3. Additionally, assurance or validation is needed to avoid potential biases or (un)fairness of the video summarization dataset when using natural watching behavior as a basis for video summarization. By addressing these concerns and providing more detailed insights, the proposed approach could be refined and more suitable for a broader range of applications. This large dataset has the potential to make a significant impact on state-of-the-art (SOTA) machine learning research. Consequently, it is crucial to ensure that the dataset is unbiased and accurately represents the video summaries in a fair manner. At present, the dataset solely relies on YouTube replay statistics without undergoing review or verification to assess its fairness or potential biases. Thorough checks may identify any potential sources of bias, such as demographic or geographic biases, in the collected statistics.__
>
> __A__. We appreciate the reviewers for raising this important question. Yes, we totally agree with the reviewer that it is important to create a fair and unbiased dataset. However, it is extremely difficult to create a large-scale completely fair and unbiased video dataset, since it is almost impossible to manually inspect all the videos and count the demographics of individuals appearing in them.
>
> As a viable remedy, we base our dataset on YouTube 8M. First of all, a YouTube video is immediately deleted if it contains any harmful content. Videos in this dataset have survived at least 7 years since 2016 when the dataset was released, and have been confirmed to be safe contents. The dataset has been created to cover a large range of topics in a balanced manner [1], which we believe is an indicator that the dataset is at least not seriously biased. In addition, the creators have hosted Kaggle competitions based on this dataset 3 times, at CVPR’17, ECCV’18, and ICCV’19, where the workshop committees carefully review the associated competition and reject if the dataset potentially causes ethical issues. The fact that they have successfully hosted the same workshops many times indicates that this dataset is sufficiently safe and reasonably unbiased (although probably no one has inspected all videos one by one).
>
> The labels we are adding are based on aggregate statistics over many users. As our ethics reviewer (8VNJ) stated, Mr. Highlight uses “widely-available public videos and metrics from YouTube in non-sensitive categories”. Additionally, “the replay information does not raise privacy concerns as it is an aggregate statistics and not attributed to any particular viewers.”
>
> [1] S. Abu-El-Haija et al., YouTube-8M: A Large-Scale Video Classification Benchmark (2016)
>
> ---
>
> __Q4. Regarding the third contribution, which involves category-specific analysis, it is not entirely clear why the specific categories of food, vehicles, and video games were chosen for the analysis. In the primary experiments, artificial scenes such as video games were excluded, and examples from sports and movies were provided. Yet, the category-specific analysis focuses on the video games category without a clear rationale. By providing a clear explanation for the selection of specific categories and their relevance to the research objectives, the paper can offer a more coherent and well-supported analysis of category-specific implications.__
>
> __A__. To make the experiment more convincing, we chose categories with a sufficient amount of videos in the dataset. Additionally, to clearly compare the difficulty level of video summarization across categories, we chose “vehicle” and “video game”, which have a wide variety of content in their videos, and “food”, which has relatively similar content in its videos.
>
> The reason for excluding artificial scenes such as video games for the primary experiment is because we had to match the data distribution to the existing ones, TVSum and SumMe, to transfer the model trained on our dataset. We mentioned this in the paper in Sec. 5.1. As TVSum and SumMe do not contain unrealistic videos like video games, we excluded it to match the data distribution.

---

### Official Review · Reviewer_CELp · 2023-07-24
**Review for Mr.Sum**

**Rating:** 7
**Confidence:** 4
**Correctness:** Claims are sound and correct.
**Clarity:** Well written.

**Strengths:**

- The proposed dataset is significantly larger than the existing ones.
- The proposed metric of most replayed statistics is intuitive and novel in the field of video summarization.
- Using replay statistics as a proxy to human labeling is great.

**Additional Feedback:**

NA

**Documentation:**

Yes

**Ethics:**

I am not sure if using replay statistics from user data can cause privacy concerns. Can the authors clarify since this is also a part of user data?

**Limitations:**

- Lacking human evaluation studies.

**Opportunities For Improvement:**

- Human evaluation should be done for a subset of videos regardless of the usage of replay statistics. Although, using replay stats is intuitive, it would be imperative to verify this for the sake of a sanity check that verifies if replay stats aligns with human annotators.

**Relation To Prior Work:**

Yes

**Summary And Contributions:**

This paper proposes a video summarization dataset that addresses the problems over existing datasets to create a consistent and meaningful summary.  The proposed dataset is called Mr. Sum and contains 31,892 videos and summarization labels aggregated over 50,000+ users per video.

---

> ### Author Response · Authors · 2023-08-24
> **Response to Reviewer CELp**
>
> We sincerely thank the reviewer for their insightful comments and suggestions. We answer each question below. Please feel free to reply to us if you have further questions.
>
> ---
>
> __Q1. Human evaluation should be done for a subset of videos regardless of the usage of replay statistics. Although, using replay stats is intuitive, it would be imperative to verify this for the sake of a sanity check that verifies if replay stats aligns with human annotators.__
>
> __A__. We thank the reviewer for this suggestion. We agree that this is a great idea, but due to the short duration of the rebuttal period, we could not conduct a user study within this period. We will add it in camera-ready.
>
> ---
>
> __Q2. I am not sure if using replay statistics from user data can cause privacy concerns. Can the authors clarify since this is also a part of user data?__
>
> __A__. As our ethics reviewer (8VNJ) stated, Mr. Highlight uses “widely-available public videos and metrics from YouTube in non-sensitive categories”. Additionally, “the replay information does not raise privacy concerns as it is an aggregate statistics and not attributed to any particular viewers.” As each statistic is aggregated with at least 50,000 viewers, it is not possible to discover viewing patterns of any particular individual.

---

### Official Review · Reviewer_gNZt · 2023-07-28
**Review for Submission315**

**Rating:** 6
**Confidence:** 4
**Correctness:** No obvious correctness errors were fo…

**Strengths:**

+ A large-scale dataset for video summarization has been presented.
+ Compared to the datasets present in the literature, Mr. Sum has a very high number of raters for each video.
+ Interesting analyzes have been conducted by the authors. The first concerns the study of the number of samples necessary to train and validate a summarization model, the second concerns the study of category-specific video summarization.

**Additional Feedback:**

Please, refer to previous sections.

**Clarity:**

The authors discussed the difference between video summarization and video highlight detection tasks. However, it is not clear how this differentiation is treated in the definition or use of the proposed dataset.

**Documentation:**

The dataset is well documented.

**Ethics:**

I don't think there are any ethical issues with the presentation.

**Limitations:**

The submission is incomplete as the authors did not include the required paper checklist.

**Opportunities For Improvement:**

- The novelty introduced by the manuscript is limited. Indeed, the authors did not contribute to the collection of the dataset, rather they very cleverly suggest how to exploit the information already available in YouTube videos. The contribution of thickness is certainly represented by the analysis part.
- From the numbers reported in Tables 4 and 5 it does not seem that Mr. Sum determines a consistent increase in performance. A gain of about 2 points in terms of F1-score is registered. The authors should better comment on this aspect.

**Relation To Prior Work:**

The authors have provided a detailed and extensive description of the literature on video summarization and related topics.

**Summary And Contributions:**

In this article, the authors presented a large-scale dataset called "Mr. Sum." This dataset comprises 31,892 videos with labels inferred from the watching behaviors of over 50,000 viewers per video. The study demonstrates that these labels effectively supervise video summarization models through transfer learning on existing benchmarks. Benchmark scores on the new dataset show improved and more stable evaluation compared to smaller benchmarks. The authors also investigated the dataset size required for robust validation and evaluation. They found that a validation set size of 1000 or larger is necessary to achieve reliable evaluation results. Additionally, a category-specific analysis using Mr. Sum revealed that the difficulty of the video summarization task varies significantly across different categories.

---

> ### Author Response · Authors · 2023-08-24
> **Response to Reviewer gNZt**
>
> We sincerely thank the reviewer for their insightful comments and suggestions. Reflecting the reviewer’s comments, we revised the manuscript accordingly (marked _blue_ in the pdf). Also, we answer each question below. Please feel free to reply to us if you have further questions.
>
> ---
>
> __Q1. The novelty introduced by the manuscript is limited. Indeed, the authors did not contribute to the collection of the dataset, rather they very cleverly suggest how to exploit the information already available in YouTube videos. The contribution of thickness is certainly represented by the analysis part.__
>
> __A__: We thank the reviewer for appreciating our analysis work. It is true that we did not recruit raters to annotate the labels, but as the reviewer admitted, we contribute the research community to use the Most replayed stats for video highlight detection and summarization at scale. Although we do not present a novel model as a regular research track NeurIPS paper, we would like to respectfully claim that providing a new dataset and labels that have not been available is also an important contribution to the community, especially because the landscape of this task is particularly noisy. This is why we submit this paper to the D&B track, where we believe these kinds of efforts are mostly welcomed.
>
> ---
>
> __Q2. From the numbers reported in Tables 4 and 5 it does not seem that Mr. Sum determines a consistent increase in performance. A gain of about 2 points in terms of F1-score is registered. The authors should better comment on this aspect.__
>
> __A__: In Table 4, SOTA models trained on our dataset not just consistently wins on the accuracy (12 categories out of 15; on average, ~2% improvement) but also they show significantly lower standard deviation (comparing Target and ZeroShot, stdev drops 1.5 to 0.4 and 1.7 to 1.2, respectively.) Table 5 also shows a similar trend, where overall performance improves 2-3% on TVSum (with comparable stdev) and significant drop on stdev (3.2 to 2.1 and 3.3 to 1.3) on SumMe (with slightly less but consistent performance improvement). We also emphasize that the improvement is achieved without changing the model, but simply training on our larger-scale dataset. To sum up, training on Mr. Highlight consistently helps the model to achieve better accuracy more robustly, although the degree somewhat depends on the target dataset.
>
> ---
>
> __Q3. The submission is incomplete as the authors did not include the required paper checklist.__
>
> __A__: We thank the reviewer for the reminder. We added the paper checklist at the end of our paper (page 11).
>
> ---
>
> __Q4. The authors discussed the difference between video summarization and video highlight detection tasks. However, it is not clear how this differentiation is treated in the definition or use of the proposed dataset.__
>
> __A__: We thank the reviewer for raising this important issue. Our main claim is that although the two tasks and the meaning of the labels are different, our Most replayed stats are suitable to supervise both tasks _without specific treatments_. We empirically verify this in Sec. 5.1. We also revised Sec. 2 with an argument that this is because the labels for the two tasks tend to be correlated, even though they are not identical.

---

### Decision · Program_Chairs · 2023-09-22

**Decision:**

Accept (Poster)

**Comment:**

The submission receives reviews from 5 different reviewers. The reviewers appreciate the scale of the dataset and acknowledge that there is no existing dataset at a similar scale for a similar problem. At the same time, multiple reviewers concern about the fact that the paper oversimplify the problem of summarization as most played or highlight. In rebuttal, the authors proposed to change it to "Mr. Highlight" to focus on highlight detection rather than summation which helps clearing concerns from reviewers. After rebuttal, all reviewers (except for reviewer 2KA9) support to accept the paper. The meta reviewer carefully read all reviews and finds the response from the authors to reviewer 2KA9 is reasonable. The meta reviewer recommends to accept the paper and request the authors to add the below items in their camera-ready to strengthen the paper.

* Requirements for camera-ready
- re-run the baselines on the entire dataset to enhance the comprehensiveness of the paper.
- human evaluation as indicated in the authors' response to reviewer CELp.
- additional baselines (e.g., CSNet) as indicated in the authors' response to reviewer 2KA9.